# Microfluidic Technologies for High Throughput Screening Through Sorting and On-Chip Culture of *C. elegans*

**DOI:** 10.3390/molecules24234292

**Published:** 2019-11-25

**Authors:** Daniel Midkiff, Adriana San-Miguel

**Affiliations:** Department of Chemical & Biomolecular Engineering, North Carolina State University, Raleigh, NC 27695, USA; dfmidkif@ncsu.edu

**Keywords:** screening, microfluidics, phenotyping, genetics, aging

## Abstract

The nematode *Caenorhabditis elegans* is a powerful model organism that has been widely used to study molecular biology, cell development, neurobiology, and aging. Despite their use for the past several decades, the conventional techniques for growth, imaging, and behavioral analysis of *C. elegans* can be cumbersome, and acquiring large data sets in a high-throughput manner can be challenging. Developments in microfluidic “lab-on-a-chip” technologies have improved studies of *C. elegans* by increasing experimental control and throughput. Microfluidic features such as on-chip control layers, immobilization channels, and chamber arrays have been incorporated to develop increasingly complex platforms that make experimental techniques more powerful. Genetic and chemical screens are performed on *C. elegans* to determine gene function and phenotypic outcomes of perturbations, to test the effect that chemicals have on health and behavior, and to find drug candidates. In this review, we will discuss microfluidic technologies that have been used to increase the throughput of genetic and chemical screens in *C. elegans*. We will discuss screens for neurobiology, aging, development, behavior, and many other biological processes. We will also discuss robotic technologies that assist in microfluidic screens, as well as alternate platforms that perform functions similar to microfluidics.

## 1. Introduction

*C. elegans* is a small, multicellular organism commonly used to model genetics, neurobiology, and other aspects of cellular and animal biology. The power of *C. elegans* as a model organism was discovered by molecular biologist Sydney Brenner in the 1960s and 1970s. Brenner found the nematode to be a strong model organism given its simple nervous system relative to the more commonly used *Drosophila melanogaster*. He was also able to use the organism as a model for genetics, and was able to identify the genetic causes for a number of visually apparent phenotypes [1].

Since the rise of *C. elegans* as a model organism, it has remained one of the most powerful systems for the study of animal biology. Despite its differences in physical structure from more complex organisms such as fish and mammals, *C. elegans* presents several advantages. First, more than 99.9% of *C. elegans* animals are self-fertilizing hermaphrodites. This unique reproductive pattern greatly simplifies genetic studies and allows for easy growth and the propagation of genetically modified or mutant lines. Second, *C. elegans* is one of the first organisms to have its entire genome sequenced [2]. Since then, a wealth of genetic information has been made available on sources such as WormBase (wormbase.org) and the NCBI (ncbi.nlm.nih.gov), maintained by the NIH and the *C. elegans* research community. Third, within wild-type strains, cellular structure is remarkably consistent. Each full-grown hermaphrodite has 959 total cells and 302 neuronal cells. Each adult cell’s origin has been tracked since the embryonic stage, allowing for relatively straightforward identification of structural defects [3]. Moreover, the structure of each neuron in the nervous system is well studied, and complete neuronal connectivity maps have been constructed for the wild-type adult [4,5]. Fourth, the *C. elegans* lifespan is approximately 2–3 weeks, which allows lifelong studies to be conducted within a month. Finally, *C. elegans* has a transparent body, which makes it ideal for in vivo imaging experiments. Fluorescent markers such as green fluorescent protein (GFP) can be used through extrachromosomal transgenes or integrated arrays to mark gene expression or protein expression [6]. All in all, *C. elegans* can be used for studying a variety of different aspects of biology with a relatively simple, easily manipulatable system [2].

Despite the potential of *C. elegans* as a model organism, conventional experimental procedures, some of which have been used since the earliest worm studies, present some limitations described below (see Section 2 for more details on traditional methods and their limitations). The application of microfluidic lab-on-a-chip technologies has increased the degree of control, accuracy, and throughput of *C. elegans* studies. Most microfluidic platforms for *C. elegans* are made from the silicone elastomer polydimethylsiloxane (PDMS) using a master mold fabricated by ultraviolet (UV) photolithography [7,8]. Devices can be fabricated at the microscale, and features can be developed down to tens of microns, ideal for use for the study and manipulation of *C. elegans*. PDMS is biocompatible and gas permeable, which also makes it possible to conduct lifelong studies of worms on-chip. Additionally, microfluidic platforms can be integrated with external hardware to temporally control and even automate experiments.

Previous reviews have given overviews of many applications of microfluidic devices for *C. elegans* studies [9,10,11], ranging from lifespan measurement on-chip [12] to assessing neuronal response to chemicals [13] and laser ablation [14]. These reviews were broad in their assessment and were not focused on a particular application of microfluidics to the study of *C. elegans*. A review written by Cornaglia et al. (2017) was more focused, as it detailed the use of microfluidics for genetic and chemical screens [15]. Since then, more devices have been developed which can be used for high-throughput screening, and will be discussed in this review. We have also included a strong emphasis on platforms for worm sorting, which are powerful tools for genetic screening. We will first review some of the microfluidic features that are commonly used in platforms for microfluidic screens. We will next provide an overview of some of the categories of high-throughput platforms which have advanced experiments in *C. elegans*. This includes platforms for sorting by worm size and developmental stage, neurobiological studies, studies of development and aging, screens for toxicology, studies of behavioral analysis, drug screening, laser ablation screening, as well as robotic technologies that can be integrated with microfluidics for high-throughput screening. This is not a comprehensive review of every platform used to perform screens, and there are other screening platforms with high impact that are not covered. We will compare and contrast platforms that perform similar functions and identify which types of studies they are most suitable for. We will conclude by assessing the overall impact of these platforms for biological research in *C. elegans.*

## 2. Traditional Methods for *C. elegans* Screening

There are two traditional methods for culture of *C. elegans*: solid agar plates or liquid media, both seeded with bacteria as a food source. Worms are either transferred in mass via a liquid suspension, or individually by either pipetting worms (in liquid media) or by picking worms using a platinum wire, both time intensive and cumbersome methods. Microscopic imaging of worms is traditionally performed by mounting animals on agarose pads on a glass slide with immobilizing drugs, such as tetramisole or levamisole. Imaging worms on a glass slide requires manual transfer to the glass slide, which is time-consuming, and immobilization with drugs that may affect gene expression or other phenotypes. Moreover, worm orientation is random when they are placed on glass slides, which makes it difficult to image worms in a specific orientation. Behavioral parameters can be measured by visual observation, such as locomotion patterns [16], or pharyngeal pumping rate [17]. These parameters can be measured without immobilization in many cases.

The ease of growth and manipulation of *C. elegans* makes them ideal for screening protocols. Screens are performed on organisms to better understand their molecular functionality and response to external stimuli. Most screens fall into the category of either chemical or genetic screens. The goal of a chemical screen is to test a collection of chemicals and identify those that produce an effect of interest. For instance, a screen might be performed for chemicals that lower or increase the survival rate of a population. This would be performed by exposing different groups of worms to different chemicals in a library and determining how survival rate compares to a control group at a certain age. Drug screens and toxicology screens are types of chemical screens that look for beneficial or damaging/lethal pharmaceutical effects, respectively. Chemical screens are often performed through use of multi-well plates. Kwok et al. screened 14,100 molecules for growth, developmental or behavioral effects by growing individual worms in 24-well plates containing solid agar and an *Escherichia coli* food source [18]. In another study, Petrascheck et al. screened 88,000 chemicals using liquid media in 384-well plates to search for lifespan-reducing chemicals [19]. Some chemical screens in liquid media make use of axenic media (free of bacteria) to avoid side effects due to *E. coli* metabolizing the added chemicals [20]. For screens of chemotaxis (the attraction or aversion of organisms to a chemical stimuli), chemical gradients must be induced by the placement of the stimuli on solid media [21]. Worm olfactory response to chemicals has also been measured by placing worms in microdroplets and exposing them to airstreams containing the odorants [22].

Genetic screens are traditionally performed in one of two manners, designated as reverse or forward genetics. Reverse genetic screens are performed by selectively knocking down gene function and assessing the phenotypic changes that occur. This is frequently achieved by performing RNA interference (RNAi) [23]. RNAi libraries contain a collection of bacterial strains, each which expresses dsRNA for one gene. Traditionally, RNAi screens are performed by picking worms into individual wells containing the RNAi food source, and then assessing their progeny for phenotypes. This requires extensive manual operation, particularly for large screens.

Alternatively, gene functionality can be determined through forward genetic methods. Through this method, worms are randomly mutagenized by exposure to a mutagen, and then individually assessed for a phenotype of interest. Putative mutants of interest are isolated, and their progeny undergo genome sequencing to identify the genetic change that causes the alternate phenotype. Simple forward genetics can be performed by visually assessing worms on a plate for variant phenotypes. However, phenotypes that are less readily apparent, either due to their subtlety or the need for high-resolution imaging, are much more difficult to identify using standard imaging methods [24].

## 3. Microfluidic Components for High-Throughput Screening

Before describing the platforms that have been used to perform high-throughput screening in *C. elegans*, we will first describe some of the innovations in microfluidic technology which have led to the development of more complex screening platforms. Screening platforms typically acquire data in series or in parallel. Acquiring data in series is done by loading worms sequentially into a device where measurements are taken for one worm at a time. For many of these screens, worm immobilization is essential to perform high-resolution imaging. A feature that has led to improvements in the capability to perform screens that involve high-resolution imaging is the microfluidic tapered channel (Figure 1a). This feature was initially developed to immobilize over one hundred worms in parallel in approximately 15 min [25]. In other devices, tapered or narrow channels have been used for the sequential imaging of worms for high-throughput screening [26,27,28]. Pressure-driven flow drives worms into the channels, providing physical immobilization aid. Complete muscular immobilization can be achieved by either immobilizing drugs, temperature control [26], or CO_2_ immobilization [29]. Other devices make use of a compression layer for immobilization, which pushes worms against the wall when the layer is pressurized [30]. Worms can also be immobilized by using vacuum suction against a wall, where by applying negative pressure to a row of parallel outlet channels, worms can be immobilized against the wall of the device [31]. Worm immobilization by these methods is reversible, as immobilization did not damage the worms, and populations of worms were able to be easily recovered (this is not true of worms mounted on a glass slide).

On-chip microfluidic valves can control the flow of animals and fluid in devices, and are effective tools for control and automation, particularly for devices where worms are loaded sequentially [8]. Rohde et al. (2007) developed one of the first microfluidic valve systems for the operation of a microfluidic screening platform (the use of this chip is described in more depth below) [31]. The valves are operated by a control layer located between the flow layer (where worms flow through the device) and the glass slide against which the device is bonded. Pressure into the control layer is actuated by off-chip pneumatic sources controlled electronically. Another valve design that only required one layer was developed by Lee et al. (2013) [28]. These valves are activated by inflating closed-end channels so that they block flow through adjacent channels on the device.

Screens are performed in parallel by maintaining multiple worms on-chip for simultaneous analysis, sometimes at multiple time points. Microfluidic chamber arrays are another feature that allows for growth, maintenance, and behavioral study of many worms simultaneously and are particularly useful for screens in parallel. Microfluidic chambers were first used for *C. elegans* by Hulme et al. (2010) to study worm development and swimming behavior throughout the lifespan of the worm (Figure 1(bi,bii)) [12]. The chambers were also supplemented with tapered imaging channels to enable measurement of higher resolution phenotypes in conjunction with worm behavior (Figure 1(biii)). Microfluidic chambers have since been used for chemical screening of dozens of worms simultaneously [32]. Chambers can also be interfaced with well plates to expose worms in different chambers to different chemicals [31].

In the following sections, we will give a detailed overview of the many devices that have been created to perform high-throughput screening and data acquisition studies in *C. elegans*. The devices will be categorized as described above and will be assessed for their advantages over traditional techniques (see Table 1 for a summary of the platforms that are reviewed).

## 4. Microfluidic Sorting of Sizes and Developmental Stages

One of the simplest microfluidic screening applications for a large number of animals is to sort based on size. From a practical standpoint, sorting based on size is useful, given the need in many studies to analyze worms based on developmental stage. This is often done by age-synchronizing populations by bleaching or periods of timed egg laying on plates [79]. Alternatively, microfluidic devices can be constructed that sort for worms based on size and age. These platforms can be used to screen for mutant worms that are unusually small or large. One of the first platforms to be developed for the age synchronization of worm populations was developed by Solvas et al. (2011) to separate animals by developmental stage [33]. Several different strategies were implemented to determine the best method for age-based sorting. These included pillars for size-based separation, ″pools” or open spaces that rely on the superior motility of larger worms, and “smart mazes” that allow throughput of larvae but prevent the passage of larger worms. It was found that the “smart maze” produced the most efficient separation at the highest throughput (Figure 2a). The architecture of the maze channels allows adult worms to flow through larger channels in the smart maze, while smaller larval worms are more likely to flow through smaller channels. The device was able to achieve an output of adult worms with only 0.2% of worms being larvae, and could operate at high throughput, with about 200–300 worms analyzed per minute.

Another platform was designed by Ai et al. (2014) which improved even further on physical channel-based filtration to separate between worms of each developmental stage [34]. Sequential microfluidic separation subunits were connected in series to separate between worms of each size. Each subunit uses an array of pillars in order to allow free flow of worms smaller than a certain size, while trapping larger individuals. Pillar spacing is varied between each chamber depending on which size is being separated. Device height is also optimized to enhance separation. Device dimensions, particularly height, will change with the flow pressure supplied to the device. Thus, device height was set so that it would be less favorable for larger, thicker worms to flow through at lower flow rates. Separation between two sizes could be achieved at an efficiency of 95% using a single device. For the most difficult separation scenario, a population of worms containing every developmental stage between L1 and adult, a separation efficiency of at least 85% was achieved for all developmental stages. This device displays a more comprehensive method of separating each developmental stage from a mixed population of worms. Other platforms have been developed to size filter worms based on developmental stage. Dong et al. (2016) developed a platform that size filtered using a control layer [35]. Pressure applied from external sources would deflect into the flow channels, limiting the size of worms that can pass through. By applying different pressures to the control layer, animals below a certain size can pass through, while larger worms will be retained upstream of the filtration features.

Aside from size-based filtration, the difference in motility of worms at different developmental stages can also be used as a separation strategy. Electrotaxis is a phenomenon where an electric field causes worms to move from the anode to the cathode, and can be used to induce worm locomotion in a particular direction [80]. Han et al. (2012) developed a platform which utilized electrotaxis to orient worm swimming [36]. Worms are then separated based on the rate at which they swim towards the cathode. In the device, larger worms achieve a higher velocity, and thus exit the flow channels sooner than smaller worms. Worms of second larval, fourth larval, and adult stages were collected at particular time points based on the rate at which they travel through the channel. Based on flow velocity, a strong separation between larval and adult stages can be achieved. Between L2 and L4, the separation is imperfect, but does produce separations of most of each stage at a critical separation time.

Rezai et al. (2012) developed a platform that used the increased resistance of narrowed channels to sort individual worms by size based on the paralysis of worms when exposed to electric fields of a higher magnitude [37]. Similar to the device described above, directional motility is induced using electrotaxis. Separation of differing worm stages is achieved by the paralysis of different worm stages under electric fields of differing intensity. Narrowed channels in the device lead to an increase in the magnitude of the electric field based on a higher voltage across the channel. As worms flow from the main chamber to the narrowed channel, they will either continue to swim through, or become paralyzed and stop swimming. Later developmental stages will become paralyzed at a lower voltage than earlier stages. This method can be used to separate two different developmental stages by setting the voltage at a level such that worms of an older stage become paralyzed as they enter the narrowed channel while allowing the younger worms to freely pass through. This method can be used to separate large populations of various developmental stages at a selectivity of 90%. It can also be used to separate older adults (four-day adults or older) or electrotaxis-defective mutants.

Wang et al. (2015) developed another platform that uses electrotaxis to size filter all worm stages simultaneously, as well as to screen for rare male worms as well as size-based mutants in a large worm population [38]. In addition to moving in the direction of the electric field, it was found that *C. elegans* moves at an angle proportional to the strength of the electric field. It was further found that in an electric field, worms of different developmental stages traveled with a different deflection angle [81]. The device design took advantage of this aspect of electrotaxis by placing the outlet chambers at different angles relative to the device inlet (Figure 2b). The electric field was optimized to produce the strongest separation between all developmental stages. At this optimal field strength, at least 82% purity of the desired developmental stage was achieved in each channel, with as high as 89% for a mix of L2 and young adult. Additionally, lon-2 long mutants were separated at a purity of 94%. For separation of males, a purity of 94% of male worms was achieved in the outermost channels. As a result, this platform can be used for the sorting of developmental stages, size, developmental mutants, and male worms.

Other platforms actively measure the size of worms that are fed through the device, as opposed to passive methods that use size filtration or differences in motility due to electrotaxis. One platform was developed by Zhu et al. (2018) which determined the size of the worm passing through the device by measuring the electrical impedance of individual worms [39]. This technique, labeled *C. elegans* microfluidic impedance cytometry, measures the dielectric properties of the worm loaded between two electrodes once an electric field is induced. These measurements are correlated to determine the worm volume, and thus the developmental stage. One worm is loaded at a time, and is sorted to one of two outlets based on impedance measurements. Identification accuracy varied between each stage, ranging from ~81% for L3 worms to over 97% for adult worms. Another platform was developed by Dong et al. (2019) which is used to sort worms based on the identification of worm size through image processing [40]. Worms are loaded into an observation chamber, and an image processing algorithm determines the worm dimensions. Worms are then sorted between two outlets. Using this platform, 90.3% accuracy was achieved for sorting young adult worms from L4 and older adult worms, while sorting 10.4 worms per minute. In future work, these platforms can be used for screens of worms of unusual size or developmental stage at a given time point.

## 5. Neurobiology Studies

One of the most advantageous aspects of *C. elegans* as a model organism is its simple nervous system, which allows for studies of connectivity, plasticity, and neurodegeneration within its 302 neurons. Since neurons are some of the smallest features within individual worms, these studies require high-resolution imaging of individual worms. As described above, there are challenges associated with high-resolution imaging on agarose pads, as worms must be manually transferred to glass slides, with little control over orientation, and are exposed to immobilizing drugs with unknown effects on some phenotypes. This includes studies of neuronal activity, neurodegeneration, and neuronal plasticity and regeneration. Acquiring electrophysiological data from the nervous system involves similar difficulties, as worms must be individually interfaced with microelectrodes, leading to a low rate of data acquisition. In this section, we will present platforms which are used to perform screens that require the acquisition of neuronal data, including neuronal structure, gene expression, calcium gradients, and electrophysiological measurements.

### 5.1. Neuronal Imaging Platforms

One of the first platforms that was developed to perform high-throughput screening of *C. elegans* was developed by Rohde et al. (2007) [31]. Single worms are captured in the device by suction from a single channel. Worms that are not captured can be re-circulated back to the inlet of the device. Next, multiple suction channels immobilize the worm in a straight position against the wall of the device. Once the worm is imaged and phenotypic data is acquired, the worm is sorted by flow directing to either a mutant outlet or a waste outlet. The flow through the device is controlled by a valve system made from an additional control layer (described above in section on microfluidic components). Worms that are sorted from the initial screen can be fed to microfluidic chambers for time-lapse imaging while fed chemical stimuli or RNAi. Hundreds of worms can be stored in chambers on one chip, where they can be imaged over time. A variation of the device was developed with an aspiration tube beneath each chamber, allowing for microscale volumes of fluid to be delivered to each chamber from multi-well plates for chemical/RNAi screening, or for worm dispensing after sorting. Thus, these devices can be used for either high-throughput sorting by phenotype, or for high throughput screens of hundreds of chemicals.

Another platform was developed for high-throughput imaging and sorting of worms by Chung et al. (2008) [26]. The platform consists of a single imaging channel that immobilizes one worm at a time via targeted temperature control. The platform is integrated with software that operates the setup automatically by detecting, imaging, and sorting worms with minimal human interaction. The platform can accurately detect subtle phenotypes, and several hundred worms can be sorted per hour, greatly reducing the time to sort worms. Phenotypes were quantified from the image using MATLAB software. The device was initially used for analyzing the autofluorescence in the worm gut, as well as for gene expression in sensory neurons. The device was also used to image fluorescently marked synapses of GABAergic motor neurons, which are about 1 micron or less in size. Thus, the immobilization is powerful enough to view high-resolution phenotypes. Use of this platform streamlines the acquisition of high-resolution imaging data and makes it much easier to recover worms after they are sorted as compared to imaging on a slide. Approximately double the number of worms that must be sorted need to be loaded into the device if either a head-first or tail-first orientation is desired. The platform was used in conjunction with a computer vision algorithm to precisely quantify phenotypes related to synapse formation. As will be described in the next paragraphs, with some modifications to the structure of this platform, a diversity of phenotypes can be analyzed and used for sorting.

An important aspect of neuronal imaging is ensuring that animals are consistently imaged in the correct orientation, so that neuronal images are taken from the same reference point. Cáceres et al. (2012) developed a platform that can be used to orient worms laterally for neuronal imaging [27]. The platform is a variant on the platform designed by Chung et al. (2008) [26] but makes use of the curved geometry to orient each worm laterally as it enters the imaging channel (Figure 3a). The lateral orientation is desirable, as it allows both the dorsal nerve cord (DNC) and ventral nerve cord (VNC) to be imaged simultaneously. A variation on the same platform was used to perform forward genetic screens for defects in synaptogenesis [82].

Lee et al. (2013) later adapted this platform to a device consisting of single PDMS layer [28]. This made the fabrication process much easier by decreasing the number of fabrication steps by eliminating the need to align two layers of PDMS. This modified version of the platform was used to screen for genes affecting expression of tryptophan hydroxylase (*tph-1)*. Approximately 4000 worms were tested for reduction in *tph-1* expression in the ADF neurons. This phenotype is of interest due to the enzymatic role of *tph-1* in the synthesis of serotonin, which modulates various bodily functions ranging from eating to mood and depression in humans. A semi-automated MATLAB algorithm was developed for device operation and phenotypic analysis and made use of an adaptive threshold for defining mutants based on data previously acquired in the dataset.

A one-layered sorting platform was later used to perform a large-scale screen for subtle mutant phenotypes that are difficult or impossible to detect with the naked eye, known as “deep phenotyping [83].” Approximately 4000 haploid genomes were sorted using machine learning algorithms to precisely quantify synaptic patterns. This allows for phenotypes based on puncta size, spacing, intensity to be quantified that are difficult or impossible to identify qualitatively or without machine vision. Using this setup, 24 mutants in synaptogenesis were identified and subsequently verified, and a new allele was found that has a function in synapse formation.

These studies above demonstrate that high-throughput imaging platforms combined with machine vision can identify phenotypes that would be impossible to detect with the naked eye at a rate that is orders of magnitude higher than what could be achieved using conventional phenotypes. A limitation of the sequential nature of these screens is that they can only reliably acquire data over a short period of time. Phenotypes that involve growth, development, and change over time cannot be screened using these platforms. Thus, these platforms are ideal for forward genetic screens which focus on phenotypes that are not time-dependent, or other sorting protocols.

Some microfluidic devices enable obtaining multiple measurements on the same worm at nearly the same time. Ma et al. (2009) developed a platform that can be used to measure the behavioral response of worms to neurotoxins while also imaging dopaminergic neurons [41]. Worms are loaded into multiple wide channels, where swimming behavior can be observed. Similar to the platform described above [26], the device makes use of a control layer to control worm loading using compression valves. In this platform the control layer expands like a balloon, pushing the worm against the side of their channel. Thus, individual worms can be repeatedly immobilized and released throughout the worm’s lifespan to both image neurons and acquire behavioral data. The device was used to test MPP+, a neurotoxin that has been shown to cause symptoms of Parkinson’s disease. Using the device, it was found that MPP+ induces motility effects and destruction of dopaminergic neurons within the worms. Thus, this platform can be used for screens and other studies that simultaneously require the measurement of locomotion and high-resolution imaging.

Other devices have been created for neuronal imaging of multiple worms simultaneously. One such device was used to measure calcium transients for up to 20 worms at a time [42]. Animals were imaged in an arena filled with micro-posts which was designed to fit within the microscopic field of view (adapted from Albrecht et al., see below [58]). Depending on the neuron of interest, calcium transients can be measured in immobilized animals or freely moving worms. A software package called NeuroTracker was used to track each worm within the chamber, collect and process fluorescent images and analyze the data to quantify behavioral responses. Using the platform, over 2800 transients were acquired from amphid sensory neurons (designated AWA) in 40 different animals exposed to different patterns of exposure to diacetyl. Additionally, over 30 different odors were tested, some which were not previously studied. The odors were delivered sequentially to the platform, and the calcium transient responses to each were recorded. As a result, a more detailed mapping of the chemical receptive field to different odors was constructed from the screening data. The platform was also used to screen for drugs which modulate neural activity.

### 5.2. Platforms for Electrophysiological Readings

While neural activity can be measured by fluorescent imaging of a calcium gradient, it can also be directly measured using electrophysiological readings of certain neurons. A device was developed by Lockery et al. (2012) to measure electrophysiolological activity of neuromuscular junctions in the *C. elegans* pharynx [43]. The device records the electropharyngeogram (EPG), or the activity of motor neurons in the pharynx, which can be measured by attaching a suction electrode to the mouth of the worm (Figure 3b). As the pharynx pumps, the large action potential of the motor neurons is detected and measured by the electrodes. Traditionally, the EPG would be measured by manually suctioning worms into a glass pipette filled with saline solution, which is used to initiate pharyngeal muscle activity. This process is time-consuming, and only a single worm can be measured at a time. The device consists of eight parallel channels with electrodes at the end to record neural activity. A population of worms is loaded into the channels via a tree-like distribution network. Chemical stimuli can then be fed to the device, where the response is measured by the electrodes in each channel. The measurements from the device successfully replicated the effects of anthelmintics (anti-nematode drugs) and was able to distinguish a drug-resistant mutant from the wild-type. In addition to anthelmintic screens, other drugs or toxins can be screened for effects on neural activity. Future iterations of the device could include on-chip reservoirs for storing multiple chemicals for delivery.

The NeuroChip was developed to sort worms based on electrophysiological readings from the pharynx [44]. The device is similar to those previously developed for worm sorting [31], but acquires electrophysiological data rather than visual data. Worms loaded into the device are oriented head-first so that the pharynx is in contact with the microelectrode. Micropillars aid in increasing the fraction of worms in the correct orientation. Signatures from distinct neural activity characteristic of mutated lines can be detected and distinguished from that of wild-type worms. The device can screen for responses at a rate of about 12 worms per hour, higher than the max of 2–4 recordings per hour that is feasible manually. Drugs and toxins may be fed to the system using a system of on-chip valves while the neural activity is being recorded. An optogenetic strain was also constructed and tested for neural activity on the chip. The platform has also been adapted for use in L2 larvae in *C. elegans*, as well as some smaller parasitic species of nematodes.

## 6. Platforms for Larval and Embryonic Development Studies

One of the key advantages of studying *C. elegans* is that the developmental progression of the nematode from the embryonic stage to adulthood is well characterized. There are four larval stages in *C. elegans* (designated L1–L4), and the cell lineage is known [3], as previously mentioned. Because worm development is so well-characterized, identification of developmental defects is relatively straightforward, making *C. elegans* a strong model organism for chemical and genetic screens for developmental alterations. Many of the methods for handling and imaging *C. elegans* adults are more challenging for embryos and larvae, as they are so small that handling of individuals by pipette or worm pick is even more difficult. In this section, we will describe platforms that can be used to perform high-throughput screens of the development of *C. elegans* larvae and embryos.

A platform called the “growthChip” was developed by Uppaluri et al. (2015) to precisely study the developmental progression of *C. elegans* larvae, and the effects of environmental conditions on worm development [45]. The device directs embryos into individual chambers using embryo traps, which catch eggs, allowing them to flow into chambers once they hatch into first stage larvae. Operation of the device is completely automated, as worms in chambers are repeatedly imaged at different time points using an automated stage. By subtracting the device background from worm images, the size of the worm is determined by a MATLAB algorithm that quantifies the worm’s area. By studying the impact of dietary regimes on rate of worm development, it was found that worms must reach a critical size before advancing to the next stage. Time spent in developmental stages could increase by orders of magnitude due to dietary restriction. It was hypothesized that increased time in achieving larval molting under dietary restriction contributed to the increased lifespan induced by dietary restriction. We anticipate that this platform can be used to perform screens on chemicals or RNAi strains that alter the time of development or alter worm size.

While this platform can be used to study the overall rate of larval development, it does not immobilize the larvae, and thus cannot be used for high-resolution imaging of larval stages. Keil et al. (2017) developed a platform to perform high-resolution imaging of larvae through the four developmental stages [30]. The platform is designed to culture worms from hatching to adulthood. Immobilization is performed completely mechanically via a second interfacial layer which compresses the growth chamber, pushing each larva to the side of the device. This allows for high resolution imaging of each animal without immobilizing drugs, which may hinder larval development. By using this platform, the cellular division sequence for development of the vulva could be imaged in detail. Both Nomarski and fluorescent microscopy could be performed, allowing for detailed studies of larval development to be conducted. High-resolution images of developmental processes such as neural branching were also acquired. Overall, this platform greatly increases the ability to study in-vivo the development of *C. elegans* larvae and can be used to screen for chemicals that affect various aspects of worm development through the larval stages.

High-resolution imaging of embryo populations is necessary to study cellular origin and gene expression patterns. A device was developed by Cornaglia et al. (2015) for long-term, high resolution imaging of embryos acquired from a population of adult worms grown on-chip (Figure 4a). This device relies on passive hydrodynamics, meaning that no on-chip valves or other active components are required [46]. The device consists of two chambers: one for on-chip culture of a stock population of adult worms used to generate the embryos for study and an embryo-incubating chamber for the culture and study of embryos (Figure 4bi–iii). *E. coli* are delivered to the device through a filter leading into the worm culture chamber. Another filter is used to transfer embryos from the main culture chamber to the embryo-incubating chamber. Individual embryos flow through the filter channels and are hydrodynamically trapped in individual micro-incubators. The flow through the channels was designed to produce stable incubation of all embryos, without any shifting or motion due to unbalanced flow. Once the embryos are in the incubator, they are imaged throughout their development time and in three dimensions. This platform can be used to perform chemical and RNAi screens for larval development through exposure of the parents, or by screening for chemicals which are absorbed by the embryos. However, multiple platforms would need to be operated in parallel to screen for multiple chemicals.

Letizia et al. (2018) upgraded this platform by adding an individual worm growth chamber to the embryo trap [47]. After the embryos hatch into larvae, the worm is thin enough to pass through a hole leading into the growth chamber. Definitive developmental markers were identified to mark transition between different larval stages. Other phenotypes that were determined were the hatching rate, average body length, vulval diameter, fertility rate, worm sex, and fluorescence-based gene expression. The device and algorithms were tested by measuring expression of *hsp-6* in response to doxycycline, a chemical that causes worm stress and slowing down development. Beyond the expected effects of doxycycline, additional, more subtle effects on developmental time and size were observed which were not previously detected. Thus, this platform can identify phenotypic changes which would have not been apparent from data sets averaged over many worms. As with the previous platform, chemical screens could potentially be performed by running multiple devices simultaneously.

A platform for both the imaging and behavior of *C. elegans* at various developmental stages was made by Atakan et al. (2019) [48]. The platform consists of eight parallel flow channels, each consisting of four worm culture chambers in series. Up to three animals could be loaded into each chamber to allow for free locomotion into adulthood. After loading, the operation of the device was automated, including automated transfer to and from the imaging channels as well as imaging and video acquisition. Flow rate leading into the imaging channels was steadily increased as worm increased in size through development. Worm length, average fluorescent intensity, and polyglutamine protein aggregation levels could all be measured in the imaging channels, while the velocity of the worm’s center could be measured in the open chamber. This platform was then used to assess a *C. elegans* model for Huntington’s disease using the polyglutamine protein Q40, and the effects of dietary restriction on disease progression. A variety of parameters, including size, aggregation level, and locomotion rate, indicated that dietary restriction had a positive impact on the health of diseased worms. Using this platform, chemical screens can be performed in the future for positive effects on disease models such as Huntington’s disease by flowing a different chemical through each of the eight inlets.

## 7. Platforms for Lifespan and Aging Studies

As mentioned above, *C. elegans* is suitable for long-term studies of lifespan and aging due to its relatively short lifespan (2–3 weeks) and its ease of genetic manipulation and analysis. Worms studied by conventional methods must be grown either on-plate or in liquid culture for the duration of each organism’s life (2–3 weeks or more). To prevent mix-up of the original generation of worms being studied with their progeny, one of two protocols must be followed. To prevent the development or hatching of embryos entirely, worms are sterilized by exposure to fluorodeoxyuridine (FUdR), or by RNAi of a gene that knocks down reproductive functions. Both approaches may affect growth, development, and aging, and may present confounding effects with various chemical screens [84]. The second option is that worm populations must remain age-synchronized throughout the egg-laying state. This is commonly done by picking worms to new plates every day or two to ensure that adults from the generation being studied do not become mixed up with progeny when these become adults. This requires many plates for each study, and time-consuming picking protocols. Even more resources are required if individual worms are to be studied longitudinally, as they must be each picked to separate plates. Additionally, there is the risk of contamination that comes with any long-term biological study. The contamination of plates with foreign bacteria and fungus can ruin experiments which have gone on for days.

Due to these challenges, there is a need to develop a more efficient and less time-consuming method for the long-term study of *C. elegans*. Thus, an effort has been made to develop microfluidic platforms for the study of individuals or populations of worms throughout their lifespan. These devices have not only increased efficiency but have also increased experimental control and reduced the external factors that may disrupt long-term studies. In this section, we will detail innovations in microfluidics which have allowed for increased efficiency in long-term worm study, as well as automated algorithms for device operation, image processing, and data analysis which increase automation and quantification of studies.

In order to perform lifespan analysis, the WormFarm platform was developed to grow populations of worms on chip throughout lifespan [49]. Eight different populations of up to 50 worms can be grown on-chip at one time, allowing data acquisition under different environmental conditions. Automated software can quantitatively determine lifespan data through video analysis and allows for unbiased data collection with less time required to count worm populations. The functionality of the platform was corroborated by testing RNAi for genes that are known to be lifespan dependent, as well as by measuring the impact of dietary regimes that affect *C. elegans* longevity. The platform is thus suitable for RNAi and drug screening for effects on lifespan. While lifespan is the most comprehensive measure of aging, there are other phenotypes that are of interest for aging studies. We will describe some of these platforms in the paragraphs below.

Doh et al. (2016) developed another microfluidic system called WormPharm (not to be confused with WormFarm described above) that can be used for lifelong studies of worm behavior [20]. Worms were grown in chambers (designated as WormChips) to which media is fed over the duration of the study. The device used an axenic medium as an alternative to conventional *E. coli* based bacterial suspensions for worm feeding. Populations of worms were imaged at time intervals throughout their lifespan. The device was used to measure the size of worm populations exposed to ethanol and nicotine at different ages compared to worms grown in media alone. Multiple WormChip platforms can be integrated into the WormPharm setup for future, larger-scale screening protocols.

As described above, devices with microfluidic chambers were created to enable the on-chip culture of *C. elegans* throughout lifespan [12]. Microfluidic platforms have since been developed to perform long term or lifelong studies of worms. Such a platform was created by Wen et al. (2012) to study the lifelong effects of stress on *C. elegans* [50]. The device consists of a circular array of 30 worm storage chambers radially connected to a single waste outlet at the center of the device, and a separate inlet flows to each chamber. A strain fluorescently expressing *gst-4* was used to screen for chemicals which increase oxidative stress throughout the worms’ lifespan. The worms were able to swim freely in individual chambers when not being imaged for *gst-4* expression levels. Through lifelong oxidative stress experiments, polydatin was found to increase stress resistance within worm populations. This platform is even more advantageous to performing high-throughput chemical screens, as each chip allows for 30 different chemicals to be fed to different chambers simultaneously.

This platform was used to further study the role of polydatin in reducing the rate of aging in *C. elegans* [85]. It was previously hypothesized that polydatin had a role in the insulin/IGF-1 signaling pathway. The platform was used to acquire lifespan, motility, and gene expression data, as well as additional data on polydatin’s increase in stress resistance. As expected, an increase in lifespan of up to 62 percent was recorded for worms under acute oxidative stress. In addition, an increase in *sod-3* expression was observed, indicating an increase in oxidative stress resistance, as well as DAF-16 localization to the nucleus, indicating an activation in response to stress. Aside from better understanding how oxidative stress can be reduced by an anti-aging compound, this study further displayed the ability to study the effects of stress on aging and worm behavior using microfluidics.

Finally, the above platform was used to study environmental effects on *C. elegans* growth, this time to study hyperglycemia which serves as a model for type 2 diabetes [86]. The effects of differing glucose concentrations were studied by measuring oxidative stress levels and lifespan, as well as lipid metabolism. The presence of glucose could reduce the lifespan of a population by 30%. Glucose also increased oxidative stress. An increased level of fat storage in worms was also observed associated with increased levels of glucose consumption. Future studies could further investigate the effects of glucose on worm health, metabolism, and gene expression, as well as beneficial impacts of drugs on worm health and lifespan. It would be of interest to see this platform used to perform more high-throughput screens of dozens of chemicals or drugs on worm behavior throughout lifespan.

Reproductive senescence is phenotype that occurs with age in *C. elegans*. Reproductive aging occurs on a much shorter timescale than other phenotypes that are indicative of age-related decline, as egg-laying typically ceases within several days of adulthood. A platform was developed by Li et al. (2015) to count the eggs laid by individual worms over time, thus making it possible to quantify reproductive senescence [51]. Individual worms were placed in separate chambers, where they were fed throughout their reproductive lifespan. As worms laid eggs, they were flushed out through the outlet of each chamber as they hatched into L1 larvae and were fed to the central hub of the device. Here, the outlet channels from each chamber converge in parallel. Images of this region are acquired at a rate of 3.75 frames per second using a LabVIEW program, ensuring that all L1 larvae are counted as they leave the device. Progeny from each chamber are continuously updated, which allows worm reproduction to be tracked in real time. The effectiveness of this platform was verified by measuring a longer reproductive lifespan in *daf-2* mutants as compared to wild-type worms. By scaling up this platform to contain more chambers, forward genetic screens could be performed for increased or decreased reproductive lifespan, or for higher or lower total brood size. The device could be interfaced with a well plate, and the progeny from each worm could be recovered once a mutant worm is identified.

Recently, a platform called the “Stress-Chip” was developed to image large worm data sets under a variety of different stressors [52]. The platform allows for precise control of growth conditions, and prevents worm escape, which frequently occurs when worms are under strenuous conditions. The device contains an open area for free worm locomotion, simulating crawling behavior with micro-posts (Figure 5B). Each of the 100 chambers on the device contains a single worm, and survival data can be acquired on-chip (Figure 5A,C). The device can be used to expose worms to a variety of different stressors, including dietary stress, oxidative stress, and osmotic stress, and can also be used to study toxicology and anti-parasitic drugs.

## 8. Platforms for Toxicity and Pathogenesis Screens

*C. elegans* can serve as an effective model organism for studying the impact of both toxic substances (toxicology) and pathogens (pathogenesis) on worm survival. Toxicology and pathogenic screens are performed in the same way that other chemical screens are performed. Typically, worm survival or other markers of health are what is studied in order to determine resistance to the pathogen or toxin, but other phenotypes may be assessed as well. In this section, we will describe microfluidic platforms that are used to streamline these types of screens.

A platform was developed by Zhang et al. (2014) to assess the toxicity of environmental manganese, particularly its effect on the degeneration of dopaminergic neurons [53]. A unique worm loading mechanism involving an electronic “worm responder” detects worms loading into the chambers and stops worm loading once the desired number of worms is loaded. The device also contains elements mixing toxicological substances with buffer solution to generate a gradient which feeds different concentrations to different chambers. Antioxidants were also fed to the device at multiple concentrations. Survival data was acquired for animals exposed different concentrations, as well as decline in fluorescently marked dopaminergic neurons and increase in oxidative stress marked by *gst-4* expression. Antioxidants were assessed for their ability to increase survival in the presence of manganese, as well as their ability to reduce oxidative stress and neuronal decline.

Many modern technologies make use of nanoparticles, particularly in the field of medicine, but there are still many questions about the toxicological impact of these particles. To answer these questions using *C. elegans* as a model organism, Kim et al. (2017) fabricated a microfluidic platform to study the uptake of silver (Ag) nanoparticles, and their effects on the health of animals [54]. Worm culture is performed on chip in chambers either exposed or not exposed to nanoparticle suspension media. Growth is studied by measuring the size of worms in tapered imaging channels. Expression patterns of metallothionein, a biomarker for toxicity in *C. elegans*, was also measured in exposed worms. Significant changes in these parameters was observed for exposure to silver nanoparticles. These effects were specific to silver, as gold nanoparticles and heavy-metal ions did not produce similar effects. This platform can be used to screen for other nanoparticles as well.

Other devices have been developed that are focused on the detrimental effects of pathogens on *C. elegans*. A platform was developed by Yang et al. (2013) to assess the effect of pathogens on populations of *C. elegans*, as well as the impact of antimicrobial effects of various compounds [55]. The device consists of 32 chambers for storing multiple worms. All chambers are loaded from the same inlet, which makes the loading process relatively fast. Once worms are loaded, pathogenic bacterial media is fed to the chambers. Compounds that are assayed for antimicrobial activity are loaded into inlets on the outside of the device. A tree-branch structure mixes with buffer, and allows a concentration gradient of the compound to be delivered to the device. Thus, different chambers were exposed to different concentrations of different compounds. Worm survival was measured for each chamber to determine the compound and concentrations that best prevent the damaging effects of the bacterial pathogen. This device could also be used to perform screens of other chemicals at a concentration gradient.

Another platform was developed by Hu et al. (2018) to study pathogenic effects and immunology over an extended period of time [56]. The focus of these studies was the progression of infections within single organisms over time. The device was also used for screening of antibiotic drugs. Individual worms were stored in separate chambers and exposed to different pathogens. Immune response was measured by the expression of the immune response gene *irg-1*, and was used to observe an increase in expression when exposed to the pathogenic bacterial strain *P. aeruginosa*. Two drugs, gentamicin and erythromycin, were tested for antimicrobial effects, and were both found to help protect the worms against the reduction in survival caused by the pathogens. Future studies using this platform could be performed to screen for larger numbers of antibiotics.

## 9. Platforms for Behavioral Screens

In the following section, we will describe platforms used to measure worm behavior, ranging from rates of locomotion to analysis of worm response to stimuli. Animals change behavior in response to food, positive/negative stimuli, heat stress, oxidative stress, and pathogens. There is a wide range of behavioral phenotypes that are of interest in *C. elegans*, ranging from locomotion rate and patterns to odor response. This section will contain a summary of a wide variety of devices that can be used to acquire large data sets from screens for behavioral analysis.

Optogenetics has emerged as a tool to study the neurobiology of *C. elegans* in recent years. The ability to induce a response to blue light stimuli in certain neurons through the genetic insertion of channelrhodopsin-2 (ChR2) into specific neurons has increased the ability to measure neural responses to a controlled, precise stimulus. A microfluidic platform was developed by Stirman et al. (2010) to screen for optogenetic response in an automated, high-throughput, and experimentally consistent manner [57]. The device consists of eight channels in parallel which straighten the worms in order to measure muscle contraction in response to optogenetic stimuli. Videos of muscle contraction were acquired automatically using LabVIEW software and processed to determine the change in worm length (contraction). The device yielded a data collection rate that was orders of magnitude higher than what was produced by conventional optogenetic methods. The platform was later used to further study the sarcomere components which regulate the contraction of muscle cells [87]. The device is also compatible with chemical stimuli and drugs, and thus can be used for screens of chemicals that affect optogenetic response.

Chemotaxis is a method used to study a population’s ability to sense and respond to either positive or negative chemical stimuli. A microfluidic device was developed by Albrecht et al. (2011) to study chemotaxis of *C. elegans* on-chip in a controlled environment [58]. The device consists of an array of microfluidic pillars in an open arena to simulate worm crawling motion. Various temporal and spatial patterns of chemical stimuli could be created to study their effects on worm chemotaxis. Temporal pulses, linear gradients, and spatial stripes could all be achieved using the device. Additionally, analysis software was developed to identify features of worm locomotion such as pauses, reversals, and pirouettes based on video footage. The device can thus be used to screen attraction or aversion to nearly any liquid-based chemical odorant.

For screens of certain phenotypes, it is important to track the behavior of individual worms over multiple time points. To accomplish this without requiring an excessive number of plates, Chung et al. (2011) developed a microfluidic chamber array containing individual worms [32]. Worms are simultaneously loaded into the chambers using a pressure-driven flow through tapered flexible loading channels, which allows for single worms to be loaded into each chamber using a syringe within minutes. This is much faster than previous techniques which require picking or pipetting worms into individual wells. The device has a low residence time, allowing different chemical stimuli to be delivered to the device at precise times. The device was used to study the impact of hermaphroditic chemical signals on the behavior of male worms. The chambers are spaced close enough that many chambers can be imaged simultaneously. This greatly speeds up the rate of data acquisition from the device.

As described above, worm locomotion is oriented based on exposure to an electric field by electrotaxis. A platform was designed by Salam et al. (2013) to simultaneously measure the behavior and neuronal structure of *C. elegans* in order to better understand the genetics that regulate the neuronal function that leads to eletrotaxis motility [59]. The platform consists of a straight channel interfaced with an anode and cathode and includes a printed scale used to quantitatively measure the motility of samples from anode to cathode (Figure 6). Using the platforms, it was found that defects in dopaminergic and sensory neurons led to defects in motility induced by an electric field. Specific defects in locomotion were measured, including reduced speed, more intermittent pauses, and abnormal turning patterns. The device was tested by exposing worms to toxic 6-hydroxydopamine (6-OHDA) and verifying that acetaminophen suppresses the negative effect of this toxin on electrotaxis. Overall, it was verified that the reduction in electrotaxis phenotype can serve as a biomarker for neuronal defects. This platform could be used for future chemical screens by feeding libraries of chemical stimuli to worms in parallel platforms.

Just as platforms have been developed to sort worms for defects in image-based phenotypes, worms can be sequentially screened for defects in worm behavior. Liu et al. (2016) developed another platform where worms can be assessed for their response to electrotaxis [60]. The platform can sort worms based on electrotactic response, and thus can be used to perform screens in an automated manner without user input. Single worms are loaded into a long channel between two electrodes using a suction channel similar to the one described previously [31] and exposed to an electric field. Electrotactic response is measured by the distance that the worm travels along the channel. The platform can be operated automatically, and screen worms for behavior at a rate of 20 worms per hour. The platform function was verified by analyzing a dopamine neurotransmitter mutant compared to a wild-type strain. The platform can be used for future chemical or genetic screens for defects in electrotaxis, which may be due to dysfunction of sensory neurons.

In addition to measuring the rate of worm locomotion, it is useful to be able to measure how chemical stimuli and genetic manipulations affect the mechanical strength of *C. elegans* body muscles. A platform was made by Johari et al. (2011) to acquire more accurate mechanical data on the force exerted by worms using an array of mechanosensory micropillars [61]. Worms are directed into three parallel channels using the valve system. Once inside, the force exertion of the worm can be measured from the horizontal displacement of the micropillars. The design of this device can be adapted to perform screens for muscular strength of individual worms or populations. It can also be used to study how chemicals or drugs impact the strength of worms. The device is effective for measuring the mechanical strength of worms, but improvements could be made to improve its effectiveness for chemical and genetic screening. By increasing the number of chambers, and by providing a separate inlet for each chamber, chemical screens can be performed for altered mechanical strength. Additionally, since on-chip valves are already used similar to those used by Chung et al. (2008) [26], two outlets for sorting can be added to the chip for sorting and forward genetic screening.

## 10. Drug Screening Platforms

Of all types of chemical screens, drug screens are one of the most important that can be performed on nematodes. The need to find faster, more efficient, and more precise methods for drug screens is amplified by both their cost and the immense need for cheaper, more effective drugs by the medical community. In the following section, we will present platforms that help achieve these goals in the earliest stage—initial drug tests on model organisms.

Carr et al. (2011) developed a drug screening platform that could be used to perform a highly sensitive dose-dependent drug screen on both *C. elegans* and other parasitic nematodes [62]. In this case, locomotion rate was the parameter used to define drug performance. This platform is unique in that it allows for observation of worms throughout the entire time that drugs are applied for a single worm. The drug that is being screened is placed in the input well which feeds into the chamber containing the worm. The chamber can be used to measure worm swimming behavior and includes a scale to precisely measure velocity. The platform and accompanying algorithm can be used to determine worm responsiveness, number of worms that leave the well, worm locomotion rate, and time until worms are unresponsive. Drug response screens can be performed at any number of three stages: pre-exposure, transient response when the drug is applied, and post-exposure behavior. The platform can also be used to perform studies on parasitic nematodes such as *O. dentatum*.

Mondal et al. (2016) designed a microfluidic device with the goal of producing a platform for high-throughput, high-resolution imaging, a goal that had previously been elusive for drug screens in *C. elegans* [63]. The device can acquire 15 z-stacks from approximately 4000 worms in 16 min. Using the platform, a total of about 100,000 worms were imaged for screening about 1000 different drugs for a reduction in levels of poly-glutamine aggregation levels. A total of four positive hits were found. The device can be used for drug screens of other phenotypes as well. Through use of this device, big data sets can be acquired in a relatively short amount of time. This platform was also used by Mondal et al. (2018) to identify reduction in age-dependent neurodegeneration [88]. A neurodegenerative model of *C. elegans* was used that is induced by a single-copy amyloid precursor protein (SC_APP) to cause deterioration in certain cholinergic neurons. A high-content screen was performed on ligands of the sigma 2 receptor, which was previously found to protect against neurodegeneration caused by the SC_APP. This was done by adding a different chemical to each well at the late larval stage, and subsequently loading the now adult worms into the imaging channels for neuronal imaging three days later. Multiple drugs were identified to have protective effects.

Ding et al. (2017) used a microfluidic chamber to test combinatorial effects of anthelmintic drugs on the survival of *C. elegans* as a model for identifying drugs that can eliminate parasitic nematodes [64]. In recent years, parasitic nematodes have become resistant to the limited number of anthelmintic drugs that are available on the market. Worms are placed in an open chamber where they are free to swim. Software measures the centroid velocity and track curvature of each worm in response to different combinations of four drugs that are fed to the device. The device relies on a feedback system control (FSC) scheme which uses iterative data from previously measured drug concentrations to modify and identify the most effective concentrations and drug combinations. In the case of anthelmintic drugs, the most effective combination yields the lowest centroid velocity with high track curvature. The FCS scheme only relies on collected data and does not require computer simulations or large amounts of data concerning the drugs being tested. This platform can be used to test combinations of drugs with other non-anthelmintic functions and can be used with parasitic nematodes as well.

Another platform was developed to perform screens for drugs that affect embryonic development, a task that is made difficult by the need to handle small embryos [65]. The device extracts embryos from egg-laying adults through use of mechanical compression, and feeds the embryos into an imaging chamber, which can image up to 100 embryos in parallel. The embryos are imaged at high-resolution at all developmental stages, from the earliest single-cell stage to a fully developed embryo right before hatching. Better control of drug concentration and temporal application can be achieved using the platform, and the device is suitable for future studies in the effects of chemicals on the earliest developmental stages of a worm’s life.

Multimodal imaging, or utilization of multiple imaging methods, can extract greater amounts of information than from one type of image acquisition. A microfluidic platform was made by Migliozzi et al. (2018) for high-throughput acquisition of multi-modal imaging data from multiple small groups of worms [66]. A series of filters on-chip is designed to separate mixed populations of worms by developmental stage. Once worms are completely immobilized, both brightfield and fluorescent images are acquired, and data is extracted. The antibiotic doxycycline was used as a proof-of-concept for the device’s ability to quantify levels of mitochondrial stress. The device was also used to assess worm motility under the influence of tetramisole as an anesthetic.

## 11. Cellular Ablation Screening Platforms

Laser ablation techniques have long been used to perform studies of cellular ablation in *C. elegans*, and have been improved to operate at the resolution of a single synapse [89]. This opened the door for studying the effects of loss of function in the *C. elegans* nervous system. As with many experimental protocols, laser ablation could be performed more successfully and efficiently if done in a high-throughput manner with minimized possibility of human error. Thus, there arose a need to perform neuronal laser ablation as well as the subsequent neuronal imaging or behavioral study on-chip. One of the first platforms used for laser ablation was fabricated by Allen et al. (2008), and allowed laser ablation to be performed on-chip, followed by imaging over time for regrowth of neurons [67]. The chip was used to study the effects of ablation of a portion of the HSNL motor neuron (which had previously been used to model synaptogenesis) at the L4 stage on development of the adult stage.

A platform was created by Guo et al. (2008) specifically to perform on-chip laser nanoaxonomy to study nerve regeneration [14]. Worms are immobilized by using a trapping membrane that is actuated by a second control layer interfaced with the flow layer. By increasing the pressure applied to the control layer, the worm in the trapping region is completely immobilized by being pushed to the edge of the channel, as well as against the glass slide. Using the platform, axonal regrowth was observed to occur much faster than previously thought. Around the same time, another platform was developed to immobilize worms for laser ablation. The platform uses an “aspiration channel” which suctions the worm against the wall of the worm flow channel, as well as a sealing membrane which inflates like a balloon, nearly completely immobilizing the worm on the side of the channel. These two elements operating together completely immobilize the worm for laser ablation. Another microfluidic device was designed to more efficiently perform cellular microsurgery by laser ablation [68]. The device is automated, and acts in a manner similar to devices used to perform high resolution imaging and sorting. A cooling channel is used to completely immobilize the worm for precise laser surgery. The platform also includes software for image processing, allowing the effects of the microsurgery to be verified. Both platforms are effective methods for immobilizing worms, however, given more recent developments in one-layer valves for the template used in the latter platform, the fabrication process might be easier for that platform.

Chemical and genetic screens can be performed on ablated worms to study neural regeneration. An automated platform was developed by Samara et al. (2010) to screen for small molecules that aid in neuronal regeneration [69]. The platform uses aspects of platforms previously developed in the Yanik lab to perform high-throughput laser ablation followed by study of neuronal regeneration [31]. A similar technique used previously to either deliver stimuli to microfluidic chambers or deliver animals to individual wells was used to repeatedly transfer worms from multi-well plates to the platform for screening. Once worms are transferred from the wells, laser ablation can be performed. Surgery could be performed on each animal in 20 s. This device was used to screen a chemical library for those which affect regeneration in the nervous system by quantifying neurite outgrowth. The screens were performed by exposing 10–20 worms to LB media containing each chemical after the worms are ablated. Worm populations were screened for an increase in neurite regrowth relative to control groups. Using this method, chemicals were identified that affect neurite outgrowth, including those that modulate protein kinase activity.

Gokce et al. (2014) developed a platform to speed up laser-based nano-surgery of axons on-chip [70] (Figure 7). Complete immobilization and accurate imaging are necessary to achieve the level of precision required for performing axonal laser ablation at the nanoscale. Worms are immobilized directly against a glass slide, which prevents refraction due to PDMS or airspace between the worm being imaged and the objective, which can lead to light diffraction and thus less accurate ablation. Once the worm is immobilized, image processing software is used to identify the precise location where laser ablation is to occur. This device can allow ablation to be performed in about 17 s per worm, which is an order of magnitude faster than manual ablation techniques. Tests comparing worms ablated on-chip vs manually with anesthetics show that there is no significant difference in the probability of neuronal reconnection. Use of this device will significantly reduce the amount of time required to screen for difference in the effects of neuronal ablation or effectiveness of reconnection mechanisms.

KillerRed is an alternate method to laser ablation which uses optogenetic activation to trigger cell death in a particular region. This method can be performed without the expensive equipment required for laser ablation; rather, a colored LED light stimulus is required. The major disadvantage of this method is the time required to perform the ablation; the ablation procedure can take between five minutes and an hour, and the time required to observe the damage can be up to 18 to 24 h. Because of this, Lee et al. (2014) developed a platform to perform optogenetic stimulation and on-chip growth of worm populations expressing KillerRed [71]. The device consists of an array of channels in parallel which can each house one worm expressing KillerRed in specific neurons. These worms are then exposed to green light, which activates the KillerRed, which produces reactive oxygen species (ROS) to destroy the neuron. The channels are wide enough to prevent clogging, and the device can operate passively to feed and maintain the worm for over 24 h. In total, 90% of worms could be recovered from the channels after the period of ablation for further study. This method is advantageous over activating KillerRed on plate, as neuronal death can be tracked in individual worms. Additionally, unlike laser ablation, a high degree of immobilization is not required to perform the procedure. Using this device, worms can undergo neuronal ablation in a high-throughput manner, and can subsequently screened for differing physical and physiological effects of damaged neurons.

## 12. Screening Platforms with Miscellaneous Applications

A unique platform was created by Ghorashian et al. (2013) to speed up the retrieval of worms from individual wells in a well plate [72]. Previous techniques involved a tube which sucked worms from the wells, which would cause bubbles that interfered with measurements acquired from the worms. Additionally, supplementary mechanical components such as moving stages were required to move the tube to each well. To fix this problem, the device interfaced directly with a well plate, connecting each to a single outlet that exits the device. The flow from each well is controlled by three-way solenoid valves activated by a control board connected to computer control software using a USB port. The device operated free of bubbles to remove 16 separate worm populations from their corresponding wells for further analysis within 4.7 s. The device could also be adapted to study larvae of zebrafish or *Drosophila*, as well as for cellular cultures, and could be used to increase the efficiency of chemical screens or other studies performed on an array of populations in a well plate.

A limitation of fabricating microfluidic devices using photolithography is the ability to fabricate features below about 1–2 microns. Even with the highest resolution photomasks available, the diffraction of UV light through the photoresist makes the fabrication of features with a high aspect ratio (i.e., small dimensions) challenging. Additionally, these tiny features can be clogged by even the tiniest pieces of debris. These challenges make it difficult to perform on-chip imaging of the smallest stage of a worm’s development—the L1 larval stage. Being the first post-embryonic stage of a developing worm, L1 imaging is necessary for studying early development and cell fate, including the transition into the dauer phase. These limitations are overcome by the development of a platform which contains immobilized L1 larvae in a thermosensitive hydrogel which will solidify at a transition temperature [73]. Worms are immobilized in gel droplets on-chip, are imaged at a high resolution, and are sorted based on a quantified phenotype. The immobilization protocol is highly reversible and causes no known effects on the development or health of larvae.

## 13. Robotics for Automated Chemical Screening

In this review, we have discussed many platforms that have been used to perform on-chip data acquisition (i.e., high-resolution imaging, swimming/crawling analysis, etc.) Other platforms are designed to greatly accelerate simpler, more traditional screening protocols. An example is the platform created by Desta et al. (2017) for speeding up loading well plates for high-throughput drug screening [74]. The device and software optically detect loaded worms. Once the worm is detected, on-chip valves release the worm, and a syringe pump pushes the worm into a microwell in a 96-well plate. A robotic arm is used to move the device outlet to individual microwells. This device serves as a rapid, completely automated method for preparing 96-well plates with individual animals for drug screens. Compared to traditional methods of worm picking or distribution by hand using pipettes, a large amount of time and effort can be saved by using this automated method.

Another platform designed to interface a microfluidic device with a well plate was developed by Lagoy et al. (2018) to streamline delivery of chemical stimuli from a 96-well plate to individual chambers [75]. Liquids are transferred to the device through a single inlet tube that retrieves chemicals from individual wells by operating a robotic arm (Figure 8). The robotic mechanism is operated by an Arduino platform, with control software developed in Arduino and Micromanager. The chemical stimuli are transferred into microfluidic arenas containing the worms to be screened. Use of this platform allows for up to 96 different chemicals to be delivered to the devices with high temporal accuracy. A screen was performed for chemicals that suppressed optogenetic neural activity. The device was also used for a ten-step protocol for cellular staining. This delivery system can be adapted to increase the throughput of a variety of chemical screens.

## 14. Non-Microfluidic Screening Methods

The COPAS CellSort is a flow cytometer which was originally created to sort large numbers of single cells in a short period of time [76]. The device has since been adapted for use with other organisms, including *C. elegans*. COPAS can sort up to 100 worms per second based on fluorescence level, size, and other visual properties. Animals are oriented laterally as they flow through the device, and points on the worm are measured longitudinally. Worms can be sorted above, below, or within a certain threshold, allowing for fast collection of worms with a specific property. While the device can be operated at a high throughput, the device itself is expensive. Moreover, only basic, large scale phenotypes can be measured, such as worm size and overall fluorescence. Thus, devices which are cheaper, more accurate, and compatible with more applications are desired.

Two methods have been developed to study the rate of food consumption in individuals or groups of *C. elegans* [77]. The first makes use of worms in a microtiter plate (a plate which can be used to measure optical density of liquid in the wells). The rate at which a worm or group of worms in a well consumes food corresponds to the rate at which the optical density of the well changes (optical density is proportional to bacterial concentration). The second method more directly measures the uptake of food by worms by feeding the worm food labeled with the stable isotope ^15^N. A higher rate of food consumption will lead to a higher level of the ^15^N isotope present in the proteins in each worm. Using the first method, it was found that the rate at which worms carrying the lifespan-extending mutation *eat-2*, which extends lifespan via dietary restriction, consumes food at a faster rate than previously thought. The second method was used to study the effects of serotonin on the rate of protein uptake. Using these methods, genetic and chemical screens can be performed for reduction in feeding rate or protein uptake. Neither of these methods are well adapted to microfluidic methods, as the two methods require microtiter plate readings or mass spectrometry, respectively. This shows that while microfluidics can improve the effectiveness of some screens, others can be better performed using alternative methods.

As an alternative to microfluidic platforms which are developed for liquid culture, the WorMotel was developed by Churgin et al. (2017) to study the lifelong progression of behavioral decline of individual worms throughout their lifespan [78]. The device consists of an array of small wells fabricated in a rectangular slab of PDMS, each containing agar with solid bacterial media. A moat containing copper sulfate solution surrounds each well, preventing worms from crawling out. Worms are manually transferred to each well. Worms can be maintained in each well throughout their lives. Along with lifespan measurements, worm locomotion was assessed by measuring the response to a flash of blue light (which induces a reflexive response in *C. elegans*). MATLAB software was developed to quantify worm locomotion based on the difference in pixel location of the worm at different time points. Each device consists of 240 wells, and 240 devices can be measured intermittently, allowing for the simultaneous study of ~57,600 worms. Using this platform, lifespan and locomotion decline were studied for wild-type and seven mutant strains, and the relations between lifespan and behavioral decline were studied. Additionally, behavioral decline under oxidative stress was compared to that with aging. Future screens can be performed for even more mutants, and both chemical and genetic screens (both forward and reverse) for alterations of lifespan can be performed using this platform.

## 15. Conclusions and Future Perspectives

As shown by the devices that we have discussed in the paragraphs above, microfluidic devices have improved the effectiveness of high-throughput screening protocols in many ways. First, microfluidic imaging channels increase the ability to perform high-resolution imaging of subtle phenotypes. The immobilization can also be aided by on-chip cooling systems, which can completely immobilize muscles quickly and reversibly. Worms can be arranged in rows for imaging in sequence, or of multiple animals at the same time. Imaging channels can supplement other elements of the devices, allowing worm images to be acquired in conjunction with behavioral data.

Second, microfluidic devices make it easier to track individual worms over a period of time through the use of chamber arrays. These chambers make it possible to monitor each individual worm over multiple time points, as opposed to collecting averaged data at each time point. For some screens, tracking individual worms can identify more subtle phenotypes that cannot be detected by an averaged data set. While individual worms can be tracked by conventional methods, it involves the use of many solid media plates, individual wells (which present their own challenges for worm imaging) or complex worm tracking software.

Third, microfluidic devices allow for greater spatial and temporal control of chemicals that can be fed to the device. Spatial gradients, for the most part, can only be created on solid media, and it is difficult for the profiles to be at the precise concentration that may be desired. Changes in chemical exposure over time can be achieved, but techniques are slower or more cumbersome. For solid media, worms must be picked to a new plate. For liquid media, samples must be centrifuged, washed, and transferred to new media. It is nearly impossible to precisely vary stimuli for worms in well plates. Microfluidic devices have a low residence time. This means that if worms are to be exposed to different stimuli, or different concentrations of stimuli, the change can be performed rapidly by switching the media that is fed to the device. Flow through microfluidic channels is laminar, and easy to predict. This makes it relatively simple to produce concentration gradients based on fluid inlets and flow rates. Creation of chemical gradients can increase the precision of gradients for study of chemotaxis studies and other chemical screens.

Fourth, integration with computer software, such as MATLAB, Arduino, and Labview can help automate protocols for device operation and data collection and analysis. This integration of hardware and software has several advantages. Operation of the devices requires less human intervention than manual studies, which can save time and effort on the part of the researcher. It also allows for more accurate, precise, and replicable protocols and measurements, and reduces the chance that human error occurs.

Fifth, microfluidic screening platforms can greatly increase the rate at which worms or chemicals are screened. Automated sorting platforms such as the one designed by Chung et al. (2008) [26] can screen for hundreds of mutant worms per hour. Devices that can perform automated screening based on laser ablation can perform the surgery an order of magnitude faster than manual techniques. Chemical and drug screens can be performed faster using devices with chamber arrays by interfacing with wells containing the chemicals to be screened [63]. Overall, automated microfluidic platforms can increase the number of mutants or chemicals to be screened due to consistent, streamlined protocols and automated analysis.

While we have noted many of the advantages of microfluidic platforms for high-throughput screening in *C. elegans*, it is also important to note some of the limitations that microfluidic devices possess. First, making microfluidic devices requires specialized equipment to perform fabrication, often done by UV photolithography. At a bare minimum, a spin coater, hot plates, and UV contact aligner are required. Particularly for devices with small features, it is necessary to have access to a cleanroom to ensure that solid particles in the air do not disrupt the substrate. UV lithography may require multiple attempts for successful fabrication. For labs that do not possess this equipment, fabrication of these devices can be difficult and expensive. Additionally, device operation can differ from lab to lab. In order to optimize fluid flow, temperature control, and other parameters, operation parameters must be tuned specifically to the lab environment. Care must be taken to inject clean media into the device, as dust and debris can clog microscale channels and disrupt device mid-operation. PDMS devices are autoclavable, but it is challenging to operate devices under completely sterile conditions. Thus, if worms are to be studied long-term after being present in the device, there may be additional risk of contamination.

Other concerns with the use of microfluidics are more specific to *C. elegans*. Using LOC technology to study yeast, embryos, or other cells is simpler relative to LOC studies on *C. elegans* given that worms move at a faster rate than the former organisms. Thus, the more unpredictable behavior of *C. elegans* can make their handling even more difficult. Since *C. elegans* has now been studied for over 50 years, many laboratories prefer to use more traditional, well-established techniques rather than investing in the new equipment and skill set required to perform microfabrication. Because of this, a select number of labs that commit to using LOC technology find microfluidic methods to be much more promising than those that don’t. As 3D printing and other technologies improve in resolution, there is hope that more labs without access to a cleanroom can make use of microfluidic technology. New commercially available chips for *C. elegans* are also expected to facilitate the adoption of this technique by the research community.

While many devices have been presented here that can be used to perform chemical and genetic screen, it would be interesting to see a higher number of studies performed using these platforms for screening. While it is helpful to hypothesize that platforms can be used for screens, as well as performing proof of concept tests or pilot screens to demonstrate the platform’s functionality, other logistical challenges can arise from performing screens at a larger scale. Logistical concerns may include the number of platforms that can be run in parallel in a lab environment, the level of operation required to maintain operation of the device for a screen, and the likelihood that operational error occurs. The larger the screen, the more likely it is that one of these factors is a concern. Additionally, certain platforms can be modified in certain ways to improve the throughput at which screens can be performed. Some platforms can be used for larger screens by increasing the number of inlets. Other platforms can be used for forward screens by adding additional sorting channels.

A major gap in much of the current microfluidic technology is the ability to recover the progeny from the worms that we have studied. As described above, microfluidic platforms that are designed for sorting make recovery of the worms that were studied simple and easy. However, these devices study worms in series rather than in parallel and cannot be used for any long-term studies. This gap is particularly challenging for forward genetic screens, which require the recovery of the progeny to identify the mutated gene. For nearly all platforms that have been reviewed, it is a tradeoff between sorting and recovering worms, and studying individual worms long-term. Platforms such as the WorMotel are a step away from achieving this ultimate goal [78]. Progeny can be recovered from worms that undergo larval developmental studies but cannot be recovered from worms that are studied for more that 2–3 days of egg laying (FuDR is required to inhibit egg-laying for such studies). A platform similar to the Chung et al. (2011) platform described above that separates eggs and larvae from individual chambers would potentially achieve this goal [32]. Perhaps the device could use a robotic mechanism similar to the one developed by Desta et al. (2017) [74].

In this review, we have presented many microfluidic platforms that have been developed within the past decade to perform high-throughput genetic and chemical screens on the nematode *C. elegans*. The devices can be used for many different types of screens, such as screens of neurons, behavioral screens, developmental and aging screens, and laser ablation screens. These devices have increased the rate at which data can be accumulated and have opened the door to more efficient and highly controlled screening experiments. Using microfluidic lab-on-a-chip technology, we anticipate an increase in the rate at which screening data can be acquired.

## Figures and Tables

**Figure 1 molecules-24-04292-f001:**
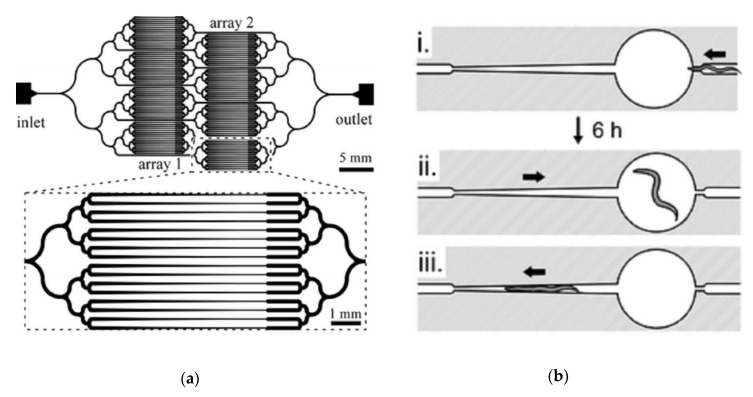
Microfluidic elements for on-chip study of *C. elegans*. (**a**) A microfluidic array of tapered channels in parallel. Worms are loaded into the device inlet and are trapped and immobilized in the tapered channels. In total, 128 parallel channels are displayed in the above image, the inset below shows 16 channels [25]. Reproduced with permission from Hulme et al., Lab. Chip; published by Royal Society of Chemistry, 2007. (**b**) Microfluidic chambers for lifelong worm culture. (**i**) Larva is loaded into chamber through inle.t (**ii**) After 6 h, the worm reaches adulthood, and is trapped in the chamber (**iii**) Inlet pressure will direct worm into channel for imaging [12]. Reproduced with permission from Hulme et al., Lab. Chip; published by Royal Society of Chemistry, 2010.

**Figure 2 molecules-24-04292-f002:**
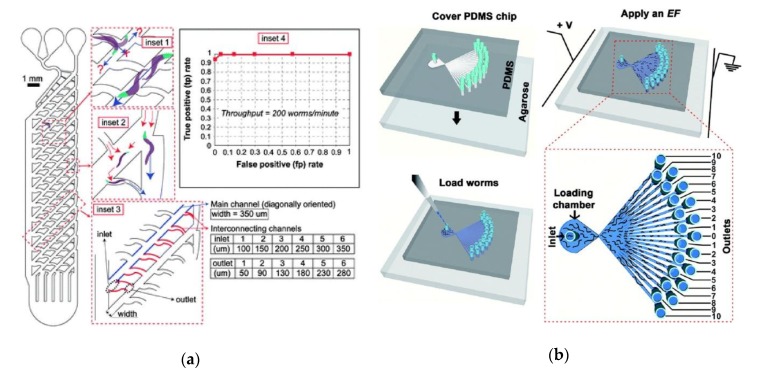
Microfluidic devices for sorting worms by developmental stage. (**a**) A “smart maze” channel structure diverts worms of different sizes to different vertical trajectories [33]. Reproduced with permission from Solvas et al., Chem. Comm.; published by Royal Society of Chemistry, 2011. (**b**) An electric field diverts worms of different sizes at different angles, allowing for size-based separation to different outlets [38]. Reproduced with permission from Wang et al., Lab. Chip; published by Royal Society of Chemistry, 2015.

**Figure 3 molecules-24-04292-f003:**
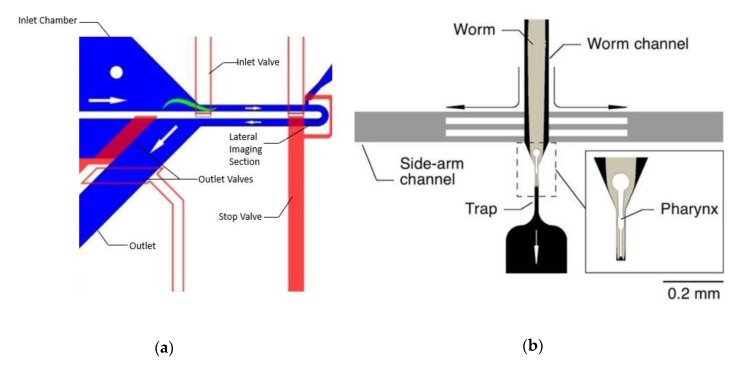
Microfluidic devices for neurobiology. (**a**) Worms flow sequentially into imaging channel for imaging in the lateral orientation [27]. Reproduced with permission from Cacéres et al., PLoS ONE.; published by Public Library of Science, 2012. (**b**) Worms are loaded and immobilized in channel for interfacing with microelectrode [43]. Reproduced with permission from Lockery et al., Lab. Chip; published by Royal Society of Chemistry, 2012.

**Figure 4 molecules-24-04292-f004:**
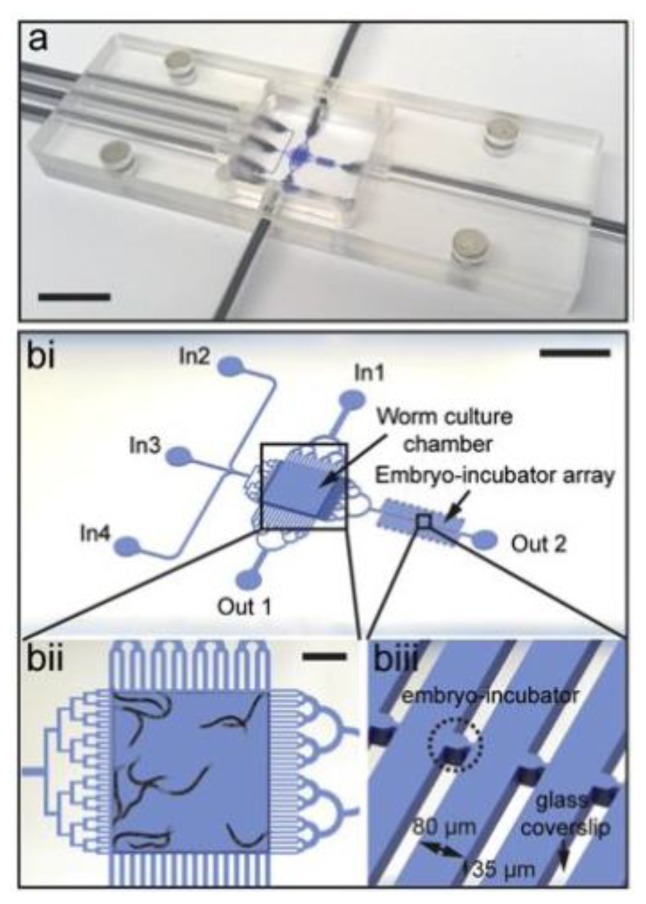
A microfluidic platform for worm culture and embryo incubation. (**a**) Actual image of device loaded with dye. (**b**) Schematic of the device. (**i**) Overall device structure including inlets, worm culture chamber, and embryo culture chamber. (**ii**) Worm culture chamber for adults. (**iii**) Incubation chamber for imaging embryonic development [46]. Reproduced with permission from Cornaglia et al., Sci. Rep., Nature Publishing Group, 2015.

**Figure 5 molecules-24-04292-f005:**
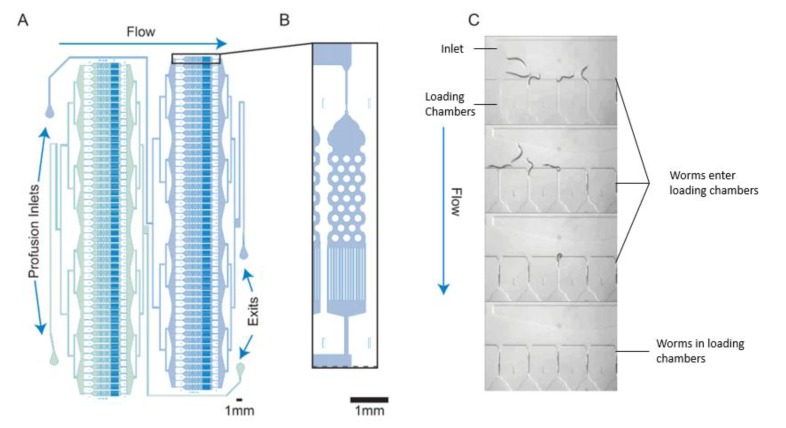
“Stress Chip” used to image *C. elegans* response to stress over lifespan. (**A**) The entire array of 100 chambers for study of individual worms. (**B**) A single chamber containing an array of micro-posts (**C**) Actual image of worms loading into chambers through inlet channels [52]. Reproduced with permission from Banse et al., PLoS ONE, Public Library of Science, 2019.

**Figure 6 molecules-24-04292-f006:**
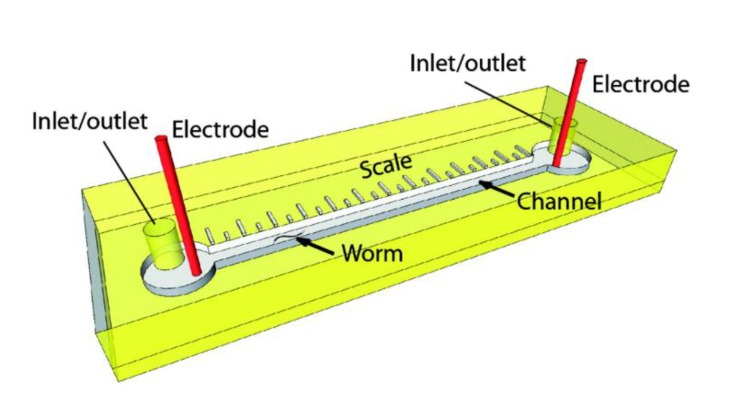
Microfluidic device for determining neuronal function by measuring electrotactic behavior. Scale quantitatively measures worm position between two electrodes [59]. Reproduced with permission from Salam et al., Worm, Taylor and Francis Online, 2013.

**Figure 7 molecules-24-04292-f007:**
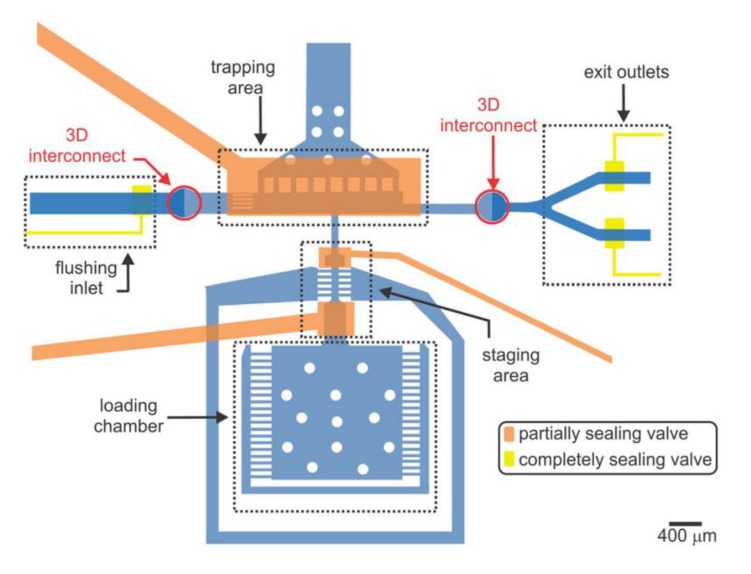
Platform developed for high-throughput laser ablation on-chip. Worms fed from the loading area through the staging area, and into the trapping area where laser ablation occurs [70]. Reproduced with permission from Gokce et al., PLoS ONE, Public Library of Science, 2015.

**Figure 8 molecules-24-04292-f008:**
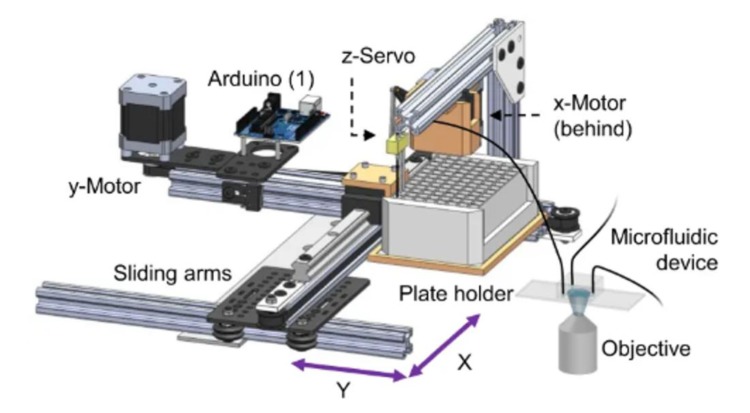
Automated robotic system for delivery of chemical stimuli from well-plates to a microfluidic device. Arduino platform controls operation of mechanism [75]. Reproduced with permission from Lagoy et al., Sci. Rep., Nature Publishing Group, 2018.

**Table 1 molecules-24-04292-t001:** Summary of Screening Platforms Reviewed.

Authors, Year	Reference	Purpose	Advantage(s)	Disadvantage(s)
*Platforms for Size-Based Sorting*				
Solvas et al. (2011)	[33]	Size-Based Sorting	Can separate larvae from adults with high accuracy and efficiency	Does not separate between differing larval stages
Ai et al. (2014)	[34]	Size-Based Sorting	Sequential separation of all larval stages at >85% efficiency	Multiple devices needed for separating all stages, leading to increased chance of operational error
Dong et al. (2016)	[35]	Size-Based Sorting	One device can separate any developmental stage based on pressure input	Throughput is 3.5 worms per second, lower than other platforms
Han et al. (2012)	[36]	Size-Based Sorting	Worms of all developmental stages can be sorted, first use of electrotaxis for size-based separation in *C. elegans*	Imperfect separation for L2–L4 stages, only 80% of worms undergo directed movement due to electrotaxis
Rezai et al. (2012)	[37]	Size-Based Sorting	Selectivity for a given developmental stage is at least 90%	Uncertain structural and molecular effects due to paralysis
Wang et al. (2015)	[38]	Size-Based Sorting	Separates all worm stages simultaneously in one device, can also isolate male worms and size mutants	Purity for certain separations as low as 82%
Zhu et al. (2018)	[39]	Size Measurement	Quantitatively measures worm size using impedance cytometry, individual worms can be sorted for forward genetic screening	Accuracy for identifying L3 worms is only 81%
Dong et al. (2019)	[40]	Size Measurement	Worm size determined using automated image analysis, individual worms can be sorted for forward genetic screens	Throughput limited to 10.4 worms per minute, clogging of channel can completely disrupt sorting
*Neurobiological Studies*				
Rohde et al. (2007)	[31]	High-Resolution Imaging and Sorting	Worms immobilized for high-resolution imaging and sorted, worms not imaged can be recycled, sorted worms can be fed to well plate	Multiple active steps required to load one worm for imaging
Chung et al. (2008)	[26]	High-Resolution Imaging and Sorting	Simplified loading step allows for fast sequential imaging	Additional microfluidic control layer must be interfaced with flow layer
Cacéres et al. (2012)	[27]	High-Resolution Imaging and Sorting	Modified channel orients worms in the lateral orientation for nerve cord imaging	Additional microfluidic control layer must be interfaced with flow layer
Lee et al. (2013)	[28]	High-Resolution Imaging and Sorting	Similar function to [26] but only consists of one microfluidic layer	Device clogging can lead to disruption in sorting
Ma et al. (2009)	[41]	Long-Term High-Resolution Imaging	Individual worms can be imaged at multiple time points throughout most of their lifespan	No straightforward method for recovery of individual worms after imaging
Larsch et al. (2013)	[42]	Calcium Transient Imaging	Calcium transients can be imaged in multiple worms	Limited to lower resolution phenotypes
Lockery et al. (2012)	[43]	Electrophysiological Measurements of Neurons	Electropharyngeograms of multiple worms can be measured simultaneously	Worms cannot be recovered after data acquisition
Hu et al. (2013)	[44]	Electrophysiological Sorting	Worms can be sorted sequentially based on electrophysiological data, three times faster than manual methods	Data acquisition on chip not fully automated
*Larval and Embryonic Development Studies*				
Uppaluri et al. (2015)	[45]	Environmental Effects on Larval Growth	Software quantitatively tracks size growth of individual worms	Only eight larvae can be tracked simultaneously on a chip
Keil et al. (2017)	[30]	High-Resolution Imaging of Larvae	Individual larvae can be imaged at high resolution	No worm outlet
Cornaglia et al. (2015)	[46]	High-Resolution Imaging of Embryos	Individual embryos maintained in incubation chambers for high-resolution imaging, operation is passive	Screening multiple chemicals would require operation of multiple platforms in parallel
Letizia et al. (2018)	[47]	High-Resolution Imaging of Embryos and Worms	Operates using same mechanisms as [46], but can image worms after each embryo hatches	Screening multiple chemicals would require operation of multiple platforms in parallel
Atakan et al. (2019)	[48]	Imaging and Behavioral Analysis of Embryonic Development	Can acquire both imaging data and locomotion rate for groups of worms	Immobilization not complete, so high-resolution imaging is not possible, only three worms can fit in each chamber for unrestricted motion
*Lifespan and Aging Studies*				
Xian et al. (2013)	[49]	Lifespan Analysis of Worm Populations with Age	Worm populations can be studied for their whole lifespan by automated analysis	Multiple chambers must be arranged in parallel for screening multiple chemicals
Doh et al. (2016)	[20]	Lifelong Behavioral Analysis Using Axenic Media	Can feed worms and determine worm size at defined intervals	Effects of axenic media on some aspects of worm biology are still unknown
Wen et al. (2012)	[50]	Lifelong Stress Studies	Worms can be imaged and stored in individual chambers over time	Multiple device layers make fabrication more challenging
Li et al. (2015)	[51]	Lifelong Reproductive Measurement	Chambers in device converge in parallel for real-time counting of progeny from each worm	Device contains many narrow regions which may increase the chance of clogging
Banse et al. (2019)	[52]	Measurement of Survival Under Stress	Large data sets (~600 worms per device) can be acquired, individual worms can be tracked over time	Chip cannot perform on-chip immobilization for high-resolution imaging
*Toxicology and Pathogenesis Studies*				
Zhang et al. (2014)	[53]	Toxic Effects on Neurons	Device inlets are mixed to create a gradient of concentrations across the device, counting mechanism loads the desired number of worms	Device must me interfaced with an electrode layer, making fabrication more complicated
Kim et al. (2017)	[54]	Toxic Effects of Nanoparticles	Tapered channels are used for immobilization, and the distance traveled along the channel is correlated with worm size	Small features could lead to device clogging
Yang et al. (2013)	[55]	Effect of Pathogens and Antimicrobials	Device inlets are mixed to create a gradient of concentrations across the device, counting mechanism loads the desired number of worms	Only four gradient mechanisms are present on each chip, limiting the number of drugs that can be screened at a time
Hu et al. (2018)	[56]	Effect of Pathogens and Antimicrobials	Worm survival can be studied for several days, device can perform high-resolution imaging	Multiple layers are required for fabrication and device assembly
*Behavioral Studies*				
Stirman et al. (2010)	[57]	Optogenetic Response	Data acquisition rate orders of magnitude higher than for data acquired on plate	Individual worms cannot be recovered
Albrecht et al. (2011)	[58]	Chemotaxis Response	Channels in device can create a variety of spatiotemporal odorant patterns	Different devices must be used for different types of spatial patterns
Chung et al. (2011)	[32]	Chemical Effects on Behavior	Worms are simultaneously loaded into chambers, making the loading process quick	Small features can lead to clogging
Salam et al. (2013)	[59]	Electrotaxis Response	Response to electrotaxis can be quantified	Needs to be scaled up to perform a large-scale screen
Liu et al. (2016)	[60]	Electrotaxis Response	Worms can be sorted based on electrotaxis response	Only 20 worms can be screened per hour
Johari et al. (2011)	[61]	Mechanical Strength Measurements	Mechanical strength can be detected by measuring the deflection of PDMS	More chambers with a separate inlet for each chamber could increase the chambers size for chemical screens
*Drug Screening Platforms*				
Carr et al. (2011)	[62]	Drug Behavioral Response	Device can be used to many behavioral parameters, individual worms are assessed for the entire period of drug application	Does not incorporate a feature for high-resolution imaging
Mondal et al. (2016)	[63]	High Resolution Imaging Drug Screens	High-resolution phenotypes can be acquired, ~4000 worms can be screened in 16 min	Imaging may be challenging with young adults or larvae
Ding et al. (2017)	[64]	Anthelmintic Drug Screens	Feedback control system automatically optimizes concentration of drugs fed to device based on previous data	Only three chambers and three drug inlets per device
Dong et al. (2018)	[65]	Embryonic Drug Screens	Worms are compressed to extract embryos for drug screening	Larvae cannot be studied after embryos hatch
Migliozzi et al. (2018)	[66]	Multimodal Imaging for Drug Screening	Large data quantities extracted from both brightfield and fluorescent images	Only three drug inlets per device
*Cellular Ablation Studies*				
Allen et al. (2008)	[67]	Neuronal Laser Ablation	Worms are immobilized on-chip in parallel for laser ablation	Worms must leave device through inlet, requires more time than for devices with inlet and outlet
Guo et al. (2008)	[14]	Neuronal Laser Ablation	Complete immobilization is achieved for precise ablation	Sorting requires more steps than other platforms
Chung et al. (2009)	[68]	Neuronal Laser Ablation	Laser ablation is more high-throughput than for previous platforms	Multiple layers are required for device assembly
Samara et al. (2010)	[69]	Laser Ablation Chemical Screen	Worms transferred from multi-well plate to channel for ablation	Operation is not fully automated
Gokce et al. (2014)	[70]	Neuronal Laser Ablation	Device operation is fully automated	Multiple layers are required for device assembly
Lee et al. (2014)	[71]	Optogenetic KillerRed Ablation	Ablation can occur in many worms simultaneously	Separate strains must be generated to ablate different cells
*Miscellaneous Applications*				
Ghorashian et al. (2013)	[72]	Well-Plate Retrieval	Recover worms from a well plate within seconds for use in a microfluidic device	Only 16 wells are interfaced with the device
Aubry et al. (2015)	[73]	High Resolution L1 Larval Imaging	Hydrogel immobilization does not require tiny features that clog easily, worms can be recovered after imaging	Hydrogel and spacing fluid are new elements that are not commonly present in microfluidic labs
*Robotics for Microfluidic Screening*				
Desta et al. (2017)	[74]	Transfer worms from device to well plate	Robotic arm automatically transfers worms from a screening platform to a well plate	Robot setup may be challenging to replicate in a different lab environment
Lagoy et al. (2018)	[75]	Deliver chemicals from well-plate to device	Robotic arm transfers specified chemicals to device for screening	More time-consuming than delivery to devices with one inlet per well
*Non-Microfluidic Screening Platforms*				
Pulak (2006) and others	[76]	COPAS Flow Cytometry Sorting	Can sort ~100 worms per second	Platform is expensive, only large-scale phenotypes can be assessed
Gomez-Amaro et al. (2015)	[77]	Measuring Food Absorption	Techniques can either measure food consumed or protein accumulation in an organism	Assessing protein accumulation requires mass-spectrometry equipment
Churgin et al. (2017)	[78]	Behavioral Decline with Age	Worms can be maintained within individual chambers on solid media, without requiring regular bacterial perfusion	Worms must be manually placed in each chamber

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
