# Peer review of "Microfluidic Technologies for High Throughput Screening Through Sorting and On-Chip Culture of C. elegans"

_molecules, 2019, doi:10.3390/molecules24234292_

Round 1

Reviewer 1 Report

Conventional techniques for growth, imaging, and behavioral analysis of a large number of C. elegans is cumbersome. Lab-on-a-chip technologies have provided the platform for carrying out these type of experiments in high-throughput manner. The authors review microfluidic-based technologies utilized for applications in high throughput sorting and screening of the C. elegans. After discussing the traditional methods of screening C. elegans, the authors review microfluidic devices developed for high-throughput sorting, neuronal imaging, larval and embryonic development studies, lifespan and aging assays, toxicity and pathogenesis screens, and behavioral screens. This study provides a good review of available microfluidic lab-on-a-chip technologies and introducing them in a well-organized format.

Major revisions:

In introduction part of this paper, authors can go through current similar reviews [9-11] and discuss their weak and strong points, especially Ref. [11] by Cornaglia et. al. titled “Microfluidic systems for high-throughput and high content screening using the nematode Caenorhabditis elegans” published in 2017. At first glance, the obvious difference between this paper and Ref. [11] is more focus on sorting feature of these high throughput technologies. Furthermore, it seems that 10-12 additional papers published from 2017 to 2019 are also discussed in this review.

Throughout the paper, among 13 sections, all sections except number 4 (4. Microfluidic Sorting of Sizes and Developmental Stages) can be categorized under high-throughput screening, while the title of the review (“Microfluidic Technologies for High Throughput Sorting and Screening of C. elegans” ) suggests something else. From the title, a reader would expect more focus on sorting, if this is what authors want to highlight. Moreover, based on author’s first sentence in section 4 “One of the simplest microfluidic screening applications for a large number of animals is to sort based on size”, sorting can be categorized as screening as well. Also, authors can discuss why sorting C. elegans using microfluidic chips are so important while commercial devices such as COPAS can easily do that.

Overall, it would be beneficial for readers if authors can clearly discuss the advantages of current review over past reviews done on this topic.

There are several other good studies that can potentially be discussed in this review. Some examples are:

1- Multi-channel device for high-density targetselective stimulation and long-term monitoring of cells and subcellular features in C. elegans

Hyewon Lee, Shin Ae Kim, Sean Coakley, Paula Mugno, Marc Hammarlund, Massimo A. Hilliardc and Hang Lu

2- Review: Microfluidic Approaches for Manipulating, Imaging, and Screening C. elegans

Bhagwati P. Gupta and Pouya Rezai

3- Automated high-content phenotyping from the first larval stage till the onset of adulthood of the nematode Caenorhabditis elegans

Huseyin Baris Atakan, Matteo Cornaglia, Laurent Mouchiroud, Johan Auwerxb and Martin A. M. Gijs

4- Li S, Stone HA, Murphy CT. 2015. A microfluidic device and automatic counting system for the study of C. elegans reproductive aging. Lab on a Chip 15(2):524-31. 

(reproductive aging was not mentioned in the section on measuring aging and lifespan but should be added)

5- Measuring Food Intake and Nutrient Absorption in Caenorhabditiselegans

Rafael L. Gomez-Amaro, Elizabeth R. Valentine, Maria Carretero, Sarah E. LeBoeuf, Sunitha Rangaraju, Caroline D. Broaddus, Gregory M. Solis, James R. Williamson, and Michael Petrascheck

Minor revisions:

It would be interesting if authors can discuss why  - after almost two decades of widespread use of PDMS chips in other applications-  microfluidic device are not being widely used in worm studies.

The title of paper in Ref. [65] should be corrected “High-Throughput Microfluidic Sorting of <i>C. elegans</i> for Automated Force Pattern Measurement”

The subsection number 9 and 13 have been repeated twice:

“9. Platforms for Behavioral Screens” and “9. Drug Screening Platforms”

“13. Non-Microfluidic Screening Platforms” and “13. Discussion and Conclusion”

Line 171: extra period. “devices can be constructed that sort for worms based on size and age. . These platforms can be used”

Reviewer 2 Report

This paper reviewed the microfluidic technologies for high throughput sorting and screening of C. elegans. The work is interesting. The following comments should be addressed before the paper can be considered further:

Line 50, examples of “conventional experimental procedures” should be given. Line 60, the authors mentioned “Previous reviews have given overviews of many applications of microfluidic devices for C. elegans studies”. Some examples should be given in this paragraph. Fig 3a & 5c, each component in the figure should be labeled. A table should be included to compare the advantages and limitations of each devices. The section “Discussion and conclusion” should be changed to “Conclusion and future perspective”
